# The mediating role of stakeholders on green banking practices and bank's performance: The case of a developing nation

Hammna Jillani◉[1]☯*, Muhammad Nawaz Chaudhry[1]☯, Hesan Zahid[2]☯, Muhammad Navid Iqbal[3]☯

1 Department of Environmental Science and Policy, Lahore School of Economics, Lahore, Pakistan,
2 Department of Commerce and Finance, Government College University Lahore, Lahore, Pakistan,
3 Faisalabad Business School, National Textile University, Faisalabad, Pakistan

☯ These authors contributed equally to this work.
* hammnajillani@gmail.com

**Data Availability Statement:** All relevant data are within the manuscript and its Supporting information files.

## Abstract

The banking sector serves as a nudge between increased financial investments and reduced environmental impacts in the modern era of sustainability thus, integrating the social, environmental, and economic dimensions. This paper aims to explore the practices and ongoing activities on account of sustainable banking which is being practised in the Pakistani Banking Sector. A mixed methods approach using a survey with a sample size of (n = 250) and in-depth interviews of (n = 25) provides significant evidence for the research. SmartPLS4.0 was used for hypotheses testing and to ascertain the path coefficient association within the constructs. This research fills the gap in existing literature by testing and implying the mediating role of Stakeholders' Influence on the relationship between Green Banking Practices and the Bank's Performance. The results of the quantitative analysis show a positive association between variables, highlighting the role of stakeholders and their need to partake efficiently, in the countrywide execution of green banking. The qualitative analysis portrays that; green banking is currently the partial focus of the banking sector in the developing economy of Pakistan whereas, approaches like financing green projects, investments in renewable energy, in-house greening of the banking sector, and provision of easy lending facilities to encourage and support environmental initiatives are some commonly practised accomplishments of the banking sector.

## 1.0 Introduction

The financial system of a country plays a very fundamental role in the economic growth of the nation. Financial development creates means for capitalization thus, causing the degradation of the environment. South Asian countries are experiencing a paradigm shift to conform to the global criteria of sustainability and shifting towards green transformation through greening the financial system [1, 2]. Over the last century, humans have put immense pressure on environmental resources. As the industries are growing, so is the stress and demand for natural

**Funding:** The author(s) received no specific funding for this work.

**Competing interests:** The authors have declared that no competing interests exist.

resources and reserves. To deal with the possible crisis before time, an effort in the form of sustainable development has been made which involves all the sectors of the economy. One emerging concept is the sustainability of the financial sector, i.e., Green Banking [3, 4]. As the world is moving towards climate change cognizance and a green growth paradigm, it is essential to integrate sustainability in the financial sector of a country. Therefore, an adequate guidance model incorporating guidelines for the financial sector needs to be followed [5]. The commercial banking sector can integrate green practices i.e., green loans and credits to achieve *"carbon neutrality"* in developing economies [6].

Sustainable Development 1972, provides the benchmarks for the incorporation of sustainable and green practices in the corporate sector. Industrial globalization is having an impact on the environment, therefore adoption of innovative practices and initiatives in a way to continue business-as-usual, achieve economic growth while minimizing environmental degradation, and be able to respond to stakeholders' pressure is the need of the hour [7]. Stakeholders can be defined as *"any group or individual that can affect or is affected by the achievements of the organization's objectives"* [8]. Stakeholders' increasing awareness can influence the adoption of green practices [9]. A firm's choice of environmental protection approach depends upon the stakeholders' interest and influence [10].

In the modern era, Stakeholder Theory has become the fundamental foundation of many firms and the corporate sector, providing ample validation regarding the capabilities of the stakeholders. Sarro et al., [11] and Ruiz et al., [12] identified that the banking sector also comprises internal and external stakeholders i.e., customers, employees, management, media, government, environment, etc. similar to the corporate sector. The interest of stakeholders' aspect is the driving force that influences green banking practices and it is essential to investigate this phenomenon in developing economies [13, 14]. The banking sector is an important stakeholder in the developing economy of Pakistan, and the implementation of green and sustainable lending practices can help in the fostering of green growth [15]. According to the Green Banking Guidelines by the State Bank of Pakistan [16], Economic benefits, Environmental Consideration, Sustainability, and Corporate Social Responsibility (CSR) are the prime factors that integrate green banking practices in the Pakistani banking sector [17]. CSR practices along with Environmental and Social Governance (ESG) have a positive and significant impact on environmental and economic performance thus ultimately promoting sustainable development in the country [18]. Rehman et al., [19] state that there is a substantial impact of available policy and guidelines in the daily operations banking sector which are sustainable in nature.

There are numerous factors that can influence the adoption of green and sustainable banking including socioeconomic, cultural, and financial dimensions. Simultaneously some significant barriers i.e., political, economic, and social factors can curtail the greening process [7]. The role of stakeholders is imperative and it accelerates the implementation of new innovative policies and guidelines [20]. Developing economies are struggling to grow and in the contemporary era, it is essential to integrate sustainable initiatives [21, 22]. Therefore, this study explores and outlines the sustainable banking framework and guidelines that currently are in practice in the developing economy of Pakistan, which can help policymakers and practitioners to identify the ways to internalize green practices. The study has the following research objectives:

1. To recognize the notable green banking practices/ initiatives taken and being followed in the Pakistani banking sector for environmental impact reduction.

2. To identify the role of Green banking guidelines on the Bank's (operational, financial, and environmental) performance.

3. To determine the role of stakeholders in the implementation of green banking guidelines.

## 2.0 Literature review

### 2.1 The banking sector and the environment

Banks are the major stakeholders in the industrial sector. Banking operations can have negative impacts on the environment directly and indirectly, but majorly through its business operations i.e., financing dirty industries. In South Asian countries, only a 1% increase in financial development can deteriorate the environment by 0.147% [23]. If the banks go green, they can play a positive role in financial stability while saving the environment which is a huge initiative towards sustainable development of any country [24].

In the current era of a rapidly changing market economy, where globalization intensified the competition between the markets, banks should play a practical and positive role in considering environmental and ecological aspects as part of their lending principle. As per the aforementioned criteria industries are supposed to use appropriate green and clean technologies, management systems, and compliance with environmental laws.

The banking sector has direct and indirect impacts on the environment and is also subjected to great stakeholders' pressure to reduce its environmental externalities [25]. Developing countries face many hurdles in the adoption and implementation phase [26] which require facilitation in order to implement green banking practices.

### 2.2 Sustainable/Green banking

Greening of the financial sector means that all investments and offering of loans should be made sustainably while considering the environmental impacts [27]. The notion of green finance comprehends that public and private sectors should establish a connection between *"technological development, innovation and the greening of the economy"* by making means for digital, online, and sustainable banking in order to discover unexploited opportunities for economic growth [28]. In the current world, the banking sector is adopting sustainable practices intending to protect the environment whilst not compromising financial performance. Green banking ideally is supposed to reduce the impacts of the financial sector on the environment [3, 4]. The Global Adaptation Index by the University of Notre Dame [29] shows that most South Asian countries are prone to climate change whereas; they lack social and environmental governance.

Over the last decade, numerous developing economies have integrated green banking into their existing banking practices and regulations. The concept of green banking links back to Dutch Origin, where environmental sustainability was incorporated into the banking sector. Later on, this concept flourished all over the world and green banking got important [30]. The Equator Principles formulated in 2003 served as the baselines for the implementation of green banking practices. The USA took the initiative to foster green banking practices and formulated the first green bank. Low-cost finance for clean/green projects, provision of great liquidity for green projects, practicing energy conservation, and financing low-carbon projects were the main aims of the green banking sector.

The banking sector has realized its accountability towards environmental protection and thus adopted the framework of Green Banking. These banking practices gained force in 2012 when the Sustainable Banking Network was developed by the International Finance Corporation, with the prime aim of green banking adoption in developing countries [31, 32].

Green banking is the merger of reduction in the operational impacts, CSR, and sustainability. The banking and financial sector should join hands to work together for sustainable development, which is the new chant [33]. Green banking is not limited to paperless/online or mobile banking; rather, it is an extensive concept that merges various dimensions to follow

sustainability in the longer run. According to IFC, green banking is defined as *"the adoption and implementation of green finance principles and practices of banks, the volume and distribution of bank assets to green investment priorities, the impacts on the quality of financial assets from integrating environmental and social factors and avoidance of negative environmental and social (E&S) impacts and the achievement of positive impacts in core financing activities"* [32].

## 2.3 Green banking in Pakistan

Green banking focuses on the responsibility of the financial sector to transform the economy making it climate resilient and having a low carbon footprint. The green banking guidelines by the State Bank of Pakistan are a nudge to a sustainable economic environment. The guidelines aim to facilitate the banking industry [16]. In the banking sector for the adoption of green banking guidelines the following parameters are to be improved: internal awareness, stakeholder engagement, financial mechanism, structural approach, and audit of the bank portfolio [20, 34]. There can be challenges in the implementation of green practices i.e., unawareness of the customers, and financial and technical obstacles [35]. Green banking serves as a paradigm shift towards sustainability. The stakeholders are the most influential factor in the adoption of green banking guidelines. Moreover, customers and users have now become aware of green banking practices and they want their banks to be more environmentally-conscious. People have become receptive to the change in conventional banking towards sustainable banking [36]. Pakistan may not be the top polluter damaging the environment but it is one of the countries that can be affected because of environmental degradation [20].

Green banking provides new investment and business opportunities through renewable energy generation projects thus, streamlining green financing for economic growth. Lending to local and corporate clients, SMEs, and farmers is to be provided as a step to promote green finance. Apart from this, an eco-friendly infrastructure needs to be developed by the banks incorporating waste, water, and energy management followed by their impact reduction targets [16].

## 2.4 Hypotheses

**2.4.1 Green banking and bank's performance.**   Shaumya and Arulrajah [37] found a significant and positive relation between green banking practices on the environmental performance of the bank in Sri Lanka, thus indicating that the performance is increased if green practices are followed. In short, all the activities should be combined to foster the environmental growth of a bank. The bank's employee-related, customer-related, operational, and policy-related activities have a direct influence on the implementation of green banking policy and green financing thus improving the environmental performance of the bank. Moreover, energy-efficient equipment and a well-designed environmental policy along with employee training and awareness sessions also contribute to the bank's environmental performance. In Bangladesh, green funding is promoted to expand the environmental performance of the banks consequently leading to the economic development of the country [38, 39]. Choudhury et al., [40] suggest that greening the banking sector and taking an environmentally proactive approach can result in functional improvements and operational efficiencies in the banking sector. Furthermore, the bank's image can be enhanced if green practices are adopted [41, 42]. The literature backs that adopting green banking practices can lead to the financial, operational, and environmental performance of the bank [43]. Henceforth, the following hypothesis is proposed.

**H1:** There is a significant relationship between Green banking practices and the Bank's performance.

**2.4.2 Stakeholders and bank's performance.** In the 20th century, stakeholders started recognizing that environmental degradation and natural resource degeneration are the greatest externalities being produced by organizations, through operation and business activities. This resulted in increasing stakeholder pressure on organizations to reduce their adverse environmental impacts. Due to the formation of environmental conferences and international protocols, excessive pressure through external groups was formed to conserve the environment and incorporate sustainability in business operations [16]. In the beginning, the banking sector was not included in the organizations that harm the environment directly and require moderation in the policies and procedures but later on, the indirect impacts of the banking sector were identified and this sustainable finance and banking came into action.

For the adoption of green banking policy, pressure from all the stakeholder groups and international organizations has a direct influence. Moreover, social pressure plays a chief role in attaining sustainable growth and development of an organization. According to the stakeholder's theory, all the groups must be equally involved, not just the financers, to make the system grow successfully and work efficiently [44, 45]. The Stakeholder Theory projects that an organization occurs for the profit and benefit of numerous stakeholders moreover it produces externalities through its business activities that can affect stakeholders [46]. As a consequence of these externalities, there is the proliferation of stakeholder pressures on firms to shrink their negative impacts. Similar to the Stakeholder Theory, the Institutional Theory also supports the stakeholder approach by arguing that creating stakeholder engagement has become essential for organizations to establish social acceptability and competitiveness along with social sustainability [47].

There exist many groups of stakeholders including media, special interest groups, employees, research community government, etc. but in the literature, four major stakeholder groups are identified that have a direct impact and influence the speeding up of the development process in any organization. The four main groups are competitors, consumers, stockholders, and top management of the organization. All the shareholders play an important role in the development of green banking procedures and activities and stakeholders should be kept on the same page through effective communication by the management of any bank [40].

According to a study by Mehedi et al., [48], the organizational pressure of various stakeholders and environmental policy has the highest influence on any organization to develop and improve sustainability in business. Shafique and Majeed [41], explored the bankers' intention to adopt green banking which can be influenced by Policy Guidelines, Attitude towards usage, Central Bank Regulations, and Management commitment and support. Linh and Anh [49], used a mixed methods approach of the questionnaire and in-depth interviews by the bank officials and the findings show the immense importance of stakeholders on green banking. Moreover, the study highlighted various benefits as green banking improves ties between local and international organizations, provides new opportunities for business growth thus minimizing capital losses, and engages all the stakeholders for organizational growth. This leads to the development of the following hypothesis.

**H2:** There is a significant relationship between Green banking practices and Stakeholders' Influence.

**H3:** Stakeholders' influence mediates the relationship between Green banking practices and the Bank's performance.

## 2.5 Theoretical framework

Bukhari et al., [42] explored the relationship among various stakeholders and their influence on greening the banking sector. There exists a positive relationship between the role of customers, competitors, and top management as influencers of green banking adoption. Moreover, the adoption of green banking practices serves as an innovation in conventional banking practices by improving banks' performance [40, 44, 49, 50].

For this research Stakeholders Theory is taken into consideration. The theory magnifies the role of internal and external stakeholders while elaborating on their ability to influence the firm. Moreover, it discusses that for a firm's performance, it is essential to keep all the stakeholders on the same page [51, 52]. Furthermore, a study by Rehman et al., [19] found an association between green banking practices and their impacts on environmental performance based on another theory called Socially Responsible Investment (SRI) which identified a strong and positive relationship between policy-related practices, daily operational practices, and green practices in Pakistan.

The research model highlights the potential relationship between the three constructs. Through this research, an attempt will be made to identify the role of Stakeholders in aiding the process of implementation of Green Banking Guidelines in the developing economy of Pakistan. Green initiatives catering to environmental protection are in the preliminary stages in various developing countries and there is a need to foster eco-friendly growth and development. This research provides on-ground practices that are being implemented at present and highlights the role of stakeholders' engagement in the effective implementation of green banking guidelines.

## 3.0 Research methodology

### 3.1 Instrument development and data collection

For this descriptive study, the data collection was completed using a purposive and snowball sampling technique, these techniques deliver in-depth perceptions and rapid data collection [53]. A structured questionnaire was used to obtain primary data from bankers from various banks [19, 37, 40, 54]. The questionnaire was developed on Google Forms and was forwarded to the bankers through the Internet, social media platforms, and bank visits. The convenience sampling technique was used to avoid time delays and ease of data entry. The survey remained active for two months and a total of n = 157 respondents filled out the form, whereas, only n = 150 were deemed fit concerning complete information. The form was again circulated to increase the number of responses and n = 250 were obtained for analysis The literature backs that an increase in sample size leads to an increased precision and lower sample error [55].

The questionnaire comprised the demographics and questions regarding the current green banking practices that are followed in various banks. Moreover, questions with respect to the Environmental, Operational, and Financial performance were also inquired. Respondents were also asked to give their opinions with respect to the role of stakeholders in the implementation of green banking guidelines. The constructs were measured with 22 indicators using a 5-point Likert Scale with 1 = 'strongly disagree' and 5 = 'strongly agree'. The questions for the survey were adapted. Moreover, a few in-depth interviews n = 25 with the bankers in senior positions were conducted to have a detailed snapshot of the ongoing banking practices and to support the questionnaire results along with the discussion.

### 3.2 Descriptive analysis

The first part of the research questionnaire provides information about the demographics. Table 1 shows the demographics of the respondents. The majority of the respondents were in

**Table 1. Descriptive profile.**

| Demographic Characteristics | Percent (n = 250) |
|---|:---:|
| **Gender** | |
| Female | 23.0 |
| Male | 77.0 |
| **Age** | |
| 18–25 | 11.3 |
| 26–35 Years | 22.7 |
| 36–45 Years | 46.0 |
| 46 Years and above | 20.0 |
| **Education** | |
| MBA | 35.3 |
| Masters/MCom | 54.7 |
| MPhil | 10.0 |
| **Designation** | |
| Banking Officer | 22.0 |
| Compliance Manager | 20.0 |
| Operations Manager | 18.0 |
| Bank Service Officer | 8.0 |
| Branch Manager | 9.0 |
| Compliance Officer | 10.8 |
| Relationship Officer | 13.0 |

the age bracket of 36 to 45 of age and the male gender was prominent with 77 percent of respondents. There is a diversity in the population sample with respect to the designation such as Compliance Managers, Operations Managers, Relationship Officers, Banking Officers, etc. are all made part of the survey to get adequate information.

## 4.0 Quantitative analysis and measurement model

The structural equation analysis of the paper was finalized using SmartPLS4.0. The research model follows the two proposed hypotheses i.e., the significant relationship between green banking practices and bank's performance and the mediating role of stakeholders on the banks' performance. The recommended analytical methods proposed by Hair et al., [56] check on the reliability and validity of the measures of the model and the structural model which is used for the Hypothesis testing were used. Moreover, following Hair et al., [56] path coefficient was analyzed. A total of 22 constructs with acceptable factor loading values were selected and the rest values were dropped to achieve good results. Composite reliability (CR) measures how well the latent variable is measured through the constructs. The CR to be reliable has to be greater than 0.7, [57]. Furthermore, to check for the construct reliability, average variance (AVE) was extracted. According to Hair et al., [58] in order for the AVE to be of acceptable level it needs to be greater than 0.5. A value greater than 0.5 shows that explained variance is greater than an unexplained variance. To measure the validity of the model Convergent and Discriminant Validity was conducted. Fig 1 gives the measurement model.

### 4.1 Convergent reliability

The convergent reliability comprises composite reliability (CR) and average variance extracted (AVE). The measurement model presents the factor loading of the chosen constructs. The

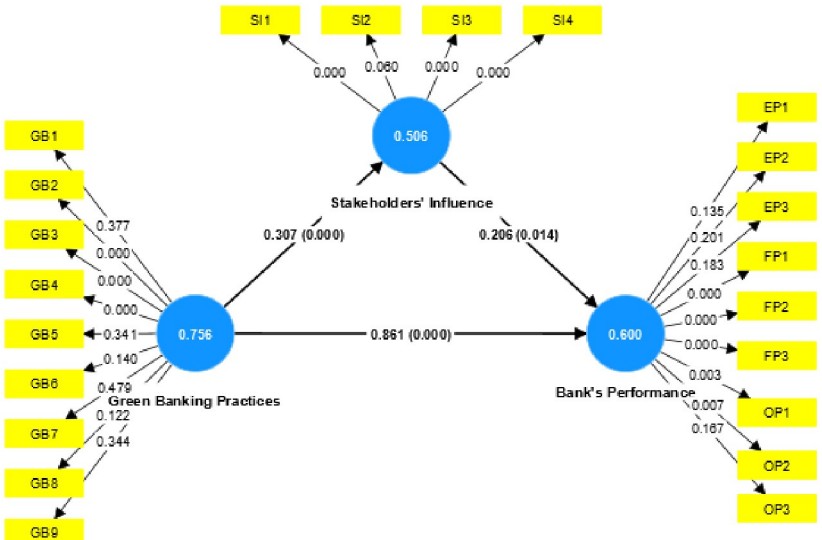

**Fig 1. Measurement model.**

items used to represent constructs pose satisfactory internal consistent relationships and a Cronbach's alpha value of i.e. >0.5 [58]. The low factor loadings collectively gave a good value of Cronbach's alpha therefore, items are not dropped. The Variance Inflation Factor (VIF) for each construct can also be seen in Table 2 which measures multicollinearity. All the values lie below 5 indicating that there is no multicollinearity among the variables.

**Table 2. Measurement model.**

| Constructs | Item no | Factor Loadings (>0.4) | Cronbach α (>0.5) | CR (>0.6) | AVE (>0.5) | VIF (<1) |
|---|---|---|---|---|---|---|
| Green Banking Practices (GB) | GB1 | 0.481 | 0.600 | 0.915 | 0.548 | 1.979 |
| | GB2 | 0.452 | | | | 1.067 |
| | GB3 | 0.756 | | | | 1.511 |
| | GB4 | 0.844 | | | | 1.623 |
| | GB5 | 0.493 | | | | 2.125 |
| | GB6 | 0.598 | | | | 2.037 |
| | GB7 | 0.449 | | | | 2.002 |
| | GB8 | 0.449 | | | | 1.350 |
| | GB9 | 0.596 | | | | 2.039 |
| Stakeholders' Influence (SI) | SI1 | 0.726 | 0.756 | 0.752 | 0.510 | 1.112 |
| | SI2 | 0.395 | | | | 1.085 |
| | SI3 | 0.581 | | | | 1.516 |
| | SI4 | 0.791 | | | | 1.707 |
| Banks' Performance (EP,FP,OP) | EP1 | 0.592 | 0.506 | 0.910 | 0.515 | 2.039 |
| | EP2 | 0.443 | | | | 1.971 |
| | EP3 | 0.531 | | | | 1.346 |
| | FP1 | 0.424 | | | | 1.965 |
| | FP2 | 0.540 | | | | 1.568 |
| | FP3 | 0.509 | | | | 1.546 |
| | OP1 | 0.431 | | | | 1.397 |
| | OP2 | 0.468 | | | | 1.537 |
| | OP3 | 0.566 | | | | 1.228 |

**Table 3. Measurement model: HTMT ratio.**

| Constructs | Heterotrait-monotrait ratio (HTMT) |
|---|---|
| Green Banking Practices <-> Bank's Performance | 0.657 |
| Stakeholders' Influence <-> Bank's Performance | 0.799 |
| Stakeholders' Influence <-> Green Banking Practices | 0.482 |

HTMT <0.9

## 4.2 Discriminant validity

Discriminant validity can be determined through the Hetro-trait Mono-trait (HTMT) ratio. This ratio measures the correlation of items across the constructs and the correlation of items within the constructs. The cut-off for the HTMT ratio is <0.9 [59]. As can easily be seen from Table 3 all values lie below 0.9, which implies that within-construct correlation is high compared to across constructs.

## 4.3 Assessment of non-linearity

Table 4 indicates the result of Ramsey's RESET to assess the non-linearity. It is indicated that there exists no partial regression of BP on SI and GB. The results of bootstrapping indicate that neither of the nonlinear effects is significant thus indicating that the linear effects model is robust.

## 4.4 Confirmatory factor analysis

The model fit is measured through the indices of discrepancy measures which are calculated in Confirmatory factor analysis. Table 5 shows the values suggested by Hu and Bentler, (1998) [60]. The results indicate a good model fit.

## 4.5 Direct relationship

In order to test the aforementioned proposed hypotheses, the path coefficient and p values were analysed. Table 6 below displays the direct relationship between the variables. Based on the significant p values confirming a significant relationship between green banking practices, stakeholders' influence, and the bank's performance, H1 and H2 are accepted.

## 4.6 Mediating relationship

The mediation impact of the constructs was calculated using the direct and indirect effects and the path values. The variables of the model show that Stakeholders' Influence acts as a mediating variable, its total effect on the Bank's performance is 0.083*** with a p-value of 0.064. The path coefficient shows a partial mediation, thus leading to the acceptance of the above mentioned hypothesis H3. The following Table 7, displays the mediating relationship.

**Table 4. Non-Linearity test.**

| Non- Linear Relationship | Coefficient | p value | f | | Ramsey's RESET |
|---|---|---|---|---|---|
| GB*GB-> SI | 0.051 | 0.387 | 0.002 | | F(3,242) = 0.97, p = 0.598 |
| GB*GB -> BP | 0.035 | 0.490 | 0.001 | | |
| SI*SI -> BP | 0.023 | 0.448 | 0.001 | | |

**Table 5. Confirmatory factor analysis.**

| Index | Recommended Value | Observed Value |
|---|---|---|
| NFI | > than 0.8 | 0.852 |
| SRMR | < than 0.8 | 0.094 |
| GFI | < than 1 | 0.903 |
| CFI | < than 1 | 0.907 |
| TLI | Should be closer to 1 | 0.85 |
| RMSEA | < than 0.10 | 0.097 |

## 5.0 Qualitative data analysis

To validate mixed methods research, qualitative research is also carried out in the form of in-person interviews. A mixed methodology is considered advantageous in a single study as it can aid in the validation of the research [61]. A total of n = 25 respondents were interviewed through structured questions till the saturation in information was achieved and sufficient data was made available to support the results of the quantitative analysis. The profile of the interviewees can be seen in Table 8.

The analysis of the interviews was done using thematic content analysis, to find out the common patterns across the data set. The results show that the banks are at a very preliminary stage to commit to sustainability and implementation of green banking practices. Green banking is currently the partial focus of the banking sector. The principal reason to incorporate banks with sustainability and environmental protection is to ensure that, no limits are surpassed and environmental deterioration can be kept under control. There are policies and targets assigned by the State Bank of Pakistan (SBP) but for the in-house execution by the banks,

**Table 6. Direct relationships hypothesis testing—SEM path analysis.**

| Direct Relationship | Estimate | SE Standard deviation (STDEV) | T statistics (|O/STDEV|) | P values |
|---|---|---|---|---|
| Green Banking Practices -> Bank's Performance | 0.861*** | 0.080 | 10.788 | 0.000 |
| Green Banking Practices -> Stakeholders' Influence | 0.307*** | 0.076 | 4.059 | 0.000 |
| Stakeholders' Influence -> Bank's Performance | 0.206*** | 0.083 | 2.467 | 0.014 |

*** p<0.01,

** p<0.05,

* p<0.1

**Table 7. Mediating relationships hypothesis testing.**

| Mediation Relationship | Estimate | SE Standard deviation (STDEV) | T statistics (|O/STDEV|) | P values |
|---|---|---|---|---|
| Green Banking Practices -> Bank's Performance | 0.903*** | 0.042 | 22.224 | 0.000 |
| Green Banking Practices -> Stakeholders' Influence | 0.342*** | 0.076 | 4.059 | 0.000 |
| Stakeholders' Influence -> Bank's Performance | 0.231*** | 0.083 | 2.467 | 0.014 |
| Green Banking Practices -> Stakeholders' Influence -> Bank's Performance | 0.083*** | 0.065 | 3.393 | 0.064 |

*** p<0.01,

** p<0.05,

* p<0.1

**Table 8. Respondents' profile.**

| Gender | Male | Female |
|---|---|---|
| | 21 | 4 |
| Designation | Branch Manager | Vice President |
| | 19 | 6 |

only a theoretical framework exists instead of appropriate applied and practical implementation. The following main themes were deduced from the collected data.

## 5.1 Stakeholders' influence

Green banking practices are highly subjective to the interest and influence of stakeholders i.e. Government and Bank Management. The primary focus of the bank's management is to engage people in the banking sector as currently a very less percentage of people use banking and own an account. Therefore, focus on green banking implementation is a secondary concern to them at the very initial stages but the Government being the regulatory body, plays a fundamental role and can influence policymaking and enforce the implementation of practices by strict laws. The majority of the respondents believe that through Government enforcement and strictness, there would be a trickle-down effect and ultimate adoption and implementation of green practices can be ensured. The current fluctuating and destabilised political condition of Pakistan can serve as a hurdle in the implementation of such practices and policies. One prominent example of stakeholders' influence in Pakistan, is the provision of solar panels at a subsidised rate of around 5–7% for customers to deal with the energy crisis and shortfall in the current era. Many banks have issued their solar financing products specifically to facilitate the general public and are themselves in the process of transformation to solar-powered branches, considering renewable energy as a step towards a green and sustainable future. This was made possible when the Government and the Bank's Management both were interested in the initiative. Stakeholders play a prime role and their engagement is essential for adopting innovative practices.

## 5.2 Environmental auditing/ EMS and green banking policy

There is a lack of proper policy on green banking practices in various banks. As per the interviewees, anything in black and white becomes mandatory to follow, green banking is at an initial stage and its adoption requires awareness. Most of the banks do not have any Environmental Auditing and Environmental Management System implemented, as bank-specific regulations lack an environmental perspective at present. There exists an external audit system by the State Bank of Pakistan, during the financing phase for the industrialists. During the phase, the environmental component is taken into consideration, and the provision of NOCs is mandatory for the project proponent to obtain funding from the bank. Respondents also mentioned that the concept of 'green washing' also prevails in the form of written policies and practices but no applied implementation. Sector-specific environmental auditing in banks needs execution to achieve green banking practices. Moreover, follow-up by SBP is supposed to play an integral role in the implementation of green banking. Again, the role of stakeholders is emphasised.

## 5.3 Training/ Green branches and go green initiatives

The majority of the banks conduct periodic training sessions for the employees and a proper module on green banking is designed as a part of their training completion requirement. The banks are focusing on "training of the trainers" so that they can move a step ahead towards the

execution of green banking. Many banks have green banking training as a part of their employee appraisal. Numerous banks are following greening initiatives and employee awareness through the celebration of specific green days, plantation drives, periodic green campaigns, etc. to promote the culture of the greening of branches.

## 5.4 Paperless/E-Banking and digital banking

Making the branches green through a special focus on paperless banking and digitalization is being used as a marketing tool. Specific software is designed for the digitalization of the banking sector and banks are adopting this transformation. Improvisation of e-banking and digital banking practices are areas of key focus for bank managers. Less printing and less paper wastage, less consumption of electricity are the few basic practices being followed in almost every bank. Many banks are going ahead and opting for paperless account opening with no manual forms, following complete digitalization and systemization. Availability of Cash Deposit Machines (CDM) is also an innovation in traditional banking followed by the digitalization of bank registers i.e. cash memos, transfer, clearing, collections register, etc. but it is at a very initial stage and only in selective banks.

## 5.5 Green practices

Internationally operating banks, for instance, *Standard Chartered Bank*s are working to reduce their carbon emissions, in 2020 the bank lowered its in-house carbon emission by 38 percent. In the case of *Muslim Commercial Bank*, Environmental Due Diligence (EnvDD) has been made part of the Bank's overall assessment. Assessment of borrower and the customer based on the Environmental Risk Rating Model to assess the environmental impacts of the business operations is made part of the banking system. The banks are also committed to sustainability through environmental and social governance (ESG), having a social and environmental policy focusing on environmental risk management and impact reduction. *Habib Bank Limited* approved the first ever 'No New Coal' policy in the commercial banking sector showing the commitment to reducing greenhouse gases. Moreover, it is the only Pakistani bank as a signatory of Green Investment Principles in China. *Bank Alfalah* has introduced a green financing product as 'Alfalah Green Energy' and has taken the initiative of formulation of an Environmental risk unit that deals with the compliance of environmental laws. In the case of *Meezan Bank*, policies on in-house environmental management and a shift to green energy have been formulated. The bank has reduced 220 Metric tons of CO2 per annum in the past. In *JS Bank*, solar panel financing solutions deployed 11.8 MW of solar power in the country. The bank is part of the Green Climate Fund by UNFCCC to finance projects to mitigate the impacts of climate change, transportation, and the provision of clean energy. The bank has partnered with the World Wind Energy Association, offering clean energy at a subsidised rate. Partnership with the local government to reduce plastic and pollution. Presence of green office certification, first commercial bank to develop and implement an Environmental Management System (EMS). In the case of *Allied Bank*, the bank has reduced electricity consumption by 15%. As a green service, the bank has launched Allied solar system financing. The following Table 9 summarises the major green practices in the Pakistani Banking Sector. The majority of the respondents mentioned the subsequent practices, which are being observed in the Pakistani Banking Sector but at a very maiden stage.

## 6 Discussion

The State Bank of Pakistan issued Green Banking Guidelines (GBG) in 2017. The main aim of green banking guidelines is the smooth flow of finances for the economic benefit of the

**Table 9. Green banking practices in Pakistani banks [62] and in-person interviews.**

| | |
|---|---|
| Compliance with green banking guidelines | Cash/cheque deposit mechanisms |
| Presence of green banking office | Green business facilitation and investment |
| Solar powered branches | Paperless banking |
| Environmental and Social Risk Management Framework | Awareness sessions, campaigns, and training |
| Energy saving guidelines | Renewable energy financing |
| Green financing | Digitalization of banking |
| Energy efficient ATMs | Reduction in carbon footprints |
| Reduction in paper consumption | Renewable energy financing |
| Energy efficient lighting and waste management practices | HSE practices |

country while considering the environment and reducing the impacts and promotion of green culture within and in the projects associated with the banks. Managing the environmental risks associated with the lender is another important task that falls in the domain of green banking. The relevant lender is supposed to provide an Environmental Improvement Plan in case any risk is associated with the project for which funding is to be done so that the environmental risk can be mitigated. GBG focuses on risk management, own impact reduction, and green business facilitation along with compliance with environmental laws.

Many developing countries are on their way to mapping down the current green banking practices so that policymakers and regulatory bodies are able to better develop the policies and guidelines for the mitigation of adverse environmental impact [54, 62]. IFC [32] shows in a survey conducted in numerous developing countries that, lack of awareness, socioeconomic barriers, measurement standards for green guidelines, and lack of stakeholders' engagement and awareness are the prime hurdles in the adoption of green banking practices. The literature depicts that developing countries are heading towards sustainable banking practices; which shall turn out as a nudge for the transformative step in the future of developing economies. Approaches like financing of green projects, investments in renewable energy, promoting the culture of green development, reduction in environmental impacts, in-house greening of the banking sector, and provision of easy lending facilities to encourage and support the environmental initiative are some commonly practiced accomplishments of the Pakistani banking sector. Pakistani banks are focusing on reducing their carbon footprint by moving towards the use of solar power in the branches and saving electricity consumption. Moreover, designated green offices and desks are introduced in the banking facility which caters to the environmental dimension. Not only this, but banks are collaborating with the Government at the local level and joining hands with an international organisation to promote the culture of green growth through various initiatives.

There are certain barriers to the implementation of green banking, customers' awareness, and low literacy rates accompanied by no access to smartphones and the internet in remote areas, barriers and challenges for the implementation of digital banking practices. Excessive in-house paper consumption and non-digitalization of other legal procedures are other obstacles, in catering to the needs of customers paperless banking cannot be fully adopted but banks are opting for paperless banking where possible. Digitalization of the banking sector can also lead to savings on operational costs. Customers' migration to the Internet and digital banking is a sensitive process, the implementation of which will be gradual and steady.

The green banking system did not only help in efficient financial performance but also helped in maintaining sustainable development worldwide. Therefore, green banking is also termed as 'sustainable and ethical banking'. Countries like India, Bangladesh, Brazil,

Columbia, and China are moving towards this sustainable shift in the banking sector. According to the Green Banking Guidelines developed by the State Bank of Pakistan, it is the responsibility of the top management to conduct sessions and training, making green policies and frameworks. Moreover, a green office manager should be assigned to deal with green projects [16]. By following green practices, not only the environment is saved from devastating projects but it can lead to multiple benefits like improved brand image and competitive advantage, especially in developing countries, it also saves the bank from the credit, reputational, and environmental risks [20, 35, 62]. Thus, the results of the study support the existing literature that the greening of the banking sector may lead to an increase in operational, financial, and environmental performance.

Financial inclusion is mandatory to promote the sustainable development of an economy [63]. Sustainable development is an emerging concept that integrates social, economic, and environmental sustainability to foster development and reduction in environmental impacts. The eco-friendly initiatives have resulted in environmental benefits along with consistent financial performance [64, 65]. The current green banking practices in the Pakistani banking sector show that; noticeable efforts are being made by the Pakistani banking sectors to promote the growth of green banking which is a foremost step towards sustainable development of the economy.

## 7 Conclusion

The Asian economy is prone to threats like population expansion, poverty, and resource utilisation; therefore, the Asian economy needs to find sustainable ways to perform efficiently in the global market. As the world is moving towards a green growth paradigm it is essential for developing economies to play their part and relevant guidelines and policies should be formulated which can strengthen the sustainability in developing countries.

Financial institutions can play a fundamental role in greening the system through the endowment of green credits and green funds for environment-friendly projects, through which sustainability can be proliferated and ensured. This research investigates the initiatives taken by the Pakistani banking sector to foster the green growth of the financial sector and to analyse the effects of green banking guidelines on the banking operations in the Pakistani banking sector. Identification of the implementation status of the green banking guidelines was done with a focus on the role of the stakeholders' influence.

Green banking has emerged in the form of innovation in conventional banking practices in order to reduce the environmental impacts thus, reducing the carbon footprint of the banking sector and increasing the bank's performance [37, 66]. This study supports the findings of existing research showing that stakeholders' influence is a primary mediating variable that can help in the implementation and fostering of green banking practices leading to improved performance of the bank. It suggests further research opportunities in the form of identification of obstacles and hurdles from stakeholders' viewpoint which results in delayed implementation of green innovations thus slowing down the transition towards a sustainable society.

## 8 Limitations and future research

The paper focuses on one developing country in the South Asian economy. An extensive study of all the developing countries can be done to map down all the green initiatives that are being taken, and ways to improvise them can be devised. Moreover, the banking sector is huge; more banks and banking personnel can be engaged in future research to dig into the 'green growth' paradigm.

## Supporting information

**S1 File. The coded data set.**
(XLSX)

## Acknowledgments

The authors acknowledge their valuable institutions.

## Author Contributions

**Conceptualization:** Hammna Jillani, Muhammad Nawaz Chaudhry, Hesan Zahid.

**Methodology:** Hammna Jillani, Hesan Zahid.

**Project administration:** Hesan Zahid.

**Supervision:** Muhammad Nawaz Chaudhry.

**Writing – original draft:** Hammna Jillani.

**Writing – review & editing:** Muhammad Navid Iqbal.

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
