## [Decision Letter · Decision Letter 0]

25 Aug 2023

PONE-D-23-17774Impact of Stakeholders' Influence on Green Banking Practices: The Case of a Developing NationPLOS ONE

Dear Dr. Jillani,

Thank you for submitting your manuscript to PLOS ONE. After careful consideration, we feel that it has merit but does not fully meet PLOS ONE’s publication criteria as it currently stands. Therefore, we invite you to submit a revised version of the manuscript that addresses the points raised during the review process.

We look forward to receiving your revised manuscript.

Kind regards,

Muhammad Hashim, PhD

Academic Editor

PLOS ONE

Journal Requirements:

Reviewers' comments:

Reviewer's Responses to Questions

**Comments to the Author**

1. Is the manuscript technically sound, and do the data support the conclusions?

Reviewer #1: Partly

Reviewer #2: Partly

2. Has the statistical analysis been performed appropriately and rigorously? 

Reviewer #1: Yes

Reviewer #2: No

3. Have the authors made all data underlying the findings in their manuscript fully available?

Reviewer #1: Yes

Reviewer #2: No

4. Is the manuscript presented in an intelligible fashion and written in standard English?

Reviewer #1: Yes

Reviewer #2: Yes

5. Review Comments to the Author

Reviewer #1: This study explains the practices and ongoing activities on account of sustainable banking which is being practiced in the Pakistani Banking Sector. The research findings have proved that green banking is currently the partial focus of the banking sector in the developing economy of Pakistan. I appreciate the opportunity to review the proposed article and hope that my consideration will help to improve the work.

1. The abstract section should clearly mention the innovation and contribution of the study. Please revise accordingly.

2. The hypotheses part should add some appropriate literature and own analysis process to support the hypothesis. The literature review section tends to describe the current research status and does not support hypotheses well.

3. The study uses SEM method is suitable for the research objective, however, add the relevance and importance of this method may be easier to read for readers. And add some explanations for indicators, such as Cronbach’s alpha means what.

4. Add some notes after tables. For example, table IV should add notes for the education variables to make is easier to understand. The authors should check all the tables and do this work.

5. Add some robust tests to ensure the results, especially to avoid the endogeneity issue.

6. References are not aligned with Journal Format. Revise it.

Reviewer #2: The article is grounded in the context of green sustainable development and employs a mixed-method approach involving surveys and in-depth interviews to investigate the practices of sustainable business implementation within the banking sector of Pakistan. The study also employs structural equation modeling and regression analysis to analyze the influence of stakeholders and subsequently explores and categorizes the current practices of green banking. The article holds a certain degree of scholarly value. However, several issues are identified:

1.The logic of the introduction and literature review is unreasonable, and there is a weak linkage between different parts of the paper. Particularly in the literature review, the author introduces four stakeholder groups, namely competitors, consumers, shareholders, and top organizational executives. While the subsequent analysis only involves the government and banks.

2. The hypotheses are given without any references or theoretical backgrounds.

3. The structural equation model that used in this paper is lack of illustrating the advantages and applicability of the model.

4. The selection of latent variables lacks a solid theoretical foundation, which may impact the credibility of the research.

5.The sample size of the paper is close to the lower limit of the SEM model’s requirements. An insufficient sample can result in biases to the research findings.

6.Only a global Cronbach’s alpha index without displaying the latent variable-specific indices can affect the data’s credibility. The presence of items with low factor loadings in the factor loading table needs to be addressed by the author to bolster the persuasiveness of the study’s outcomes.

7. The RMSEA value is approaching 0.1, whereas values below 0.05 are regarded as more ideal.

8.The robustness test is absent, has the author considered incorporating robustness tests to enhance the study’s reliability?

Above all, many parts of this paper are lack of detailed and necessary discussion, it is recommended to reject the paper.

6. PLOS authors have the option to publish the peer review history of their article (what does this mean?). If published, this will include your full peer review and any attached files.

Reviewer #1: No

Reviewer #2: No

---

## [Author Response · Author response to Decision Letter 0]

21 Sep 2023

Dear Editor,

PLOS One.

Subject: Rebuttal Letter

Please be informed that the authors acknowledge the time-taking and guiding suggestions of the reviewers and the editor. We believe that the valuable comments can make the paper more effective in terms of publication and readiness. We have meticulously reviewed and addressed each comment, and all modifications have been highlighted in the revised manuscript for ease of reference. We humbly request your reconsideration of our work and eagerly await your further feedback.The required documents are attached to the submission. 

1.0 Journal Requirements

Authors’ Reply: The manuscript has been updated according to PLOS ONE’s style requirements.

Authors’ Reply: The data set is made available. The revision files will contain a supporting document of data set.

2.0 Reviewer #1

Authors’ Reply: The abstract has been revised carefully incorporating all the essential information with respect to the updated methodology and results.

Reviewer #1: This study explains the practices and ongoing activities on account of sustainable banking which is being practiced in the Pakistani Banking Sector. The research findings have proved that green banking is currently the partial focus of the banking sector in the developing economy of Pakistan. I appreciate the opportunity to review the proposed article and hope that my consideration will help to improve the work.

1. The abstract section should clearly mention the innovation and contribution of the study. Please revise accordingly.

Authors’ Reply: The literature section has been substantially revised and rewritten. The literature contains specific section with support the proposed hypothesis. 

2. The hypotheses part should add some appropriate literature and own analysis process to support the hypothesis. The literature review section tends to describe the current research status and does not support hypotheses well.

Authors’ Reply: The methodology section has been upgraded. The SEM analysis has been carried out on SmartPLS thus, improving the nature of analysis and writing.

3. The study uses SEM method is suitable for the research objective, however, add the relevance and importance of this method may be easier to read for readers. And add some explanations for indicators, such as Cronbach’s alpha means what.

Authors’ Reply: The tables have been revised as per the new methodology adopted and the detailed explanation has been provided in the subsequent paragraph.

4. Add some notes after tables. For example, table IV should add notes for the education variables to make is easier to understand. The authors should check all the tables and do this work.

Authors’ Reply: The statistical analysis has been updated using SEM-PLS. Following tests are used: convergent reliability, discriminate validity, direct relationship and mediating relationship. The hypotheses have been tested using path coefficient and p values.

5. Add some robust tests to ensure the results, especially to avoid the endogeneity issue.

6. References are not aligned with Journal Format. Revise it.

Authors’ Reply: The manuscript has been revised using journal’s guidelines.

3.0 Reviewer #2

Reviewer #2: The article is grounded in the context of green sustainable development and employs a mixed-method approach involving surveys and in-depth interviews to investigate the practices of sustainable business implementation within the banking sector of Pakistan. The study also employs structural equation modeling and regression analysis to analyze the influence of stakeholders and subsequently explores and categorizes the current practices of green banking. The article holds a certain degree of scholarly value. However, several issues are identified:

1.The logic of the introduction and literature review is unreasonable, and there is a weak linkage between different parts of the paper. Particularly in the literature review, the author introduces four stakeholder groups, namely competitors, consumers, shareholders, and top organizational executives. While the subsequent analysis only involves the government and banks.

Authors’ Reply: The introduction and the literature review have been substantially revised and rewritten. A strong link has been developed between various sections. The analysis part has also been revised and upgraded.

2. The hypotheses are given without any references or theoretical backgrounds.

Authors’ Reply: After considering the available literature the hypotheses have been developed with references. A clear theoretical framework has been provided in the manuscript.

3. The structural equation model used in this paper is lack of illustrating the advantages and applicability of the model.

Authors’ Reply: The methodology part has been updated and the analysis is upgraded using SEM through SmartPLS. Integrating this technique in the paper makes it a sound piece of research. 

4. The selection of latent variables lacks a solid theoretical foundation, which may impact the credibility of the research.

Authors’ Reply: The theoretical foundation has been significantly developed in the manuscript. 

5.The sample size of the paper is close to the lower limit of the SEM model’s requirements. An insufficient sample can result in biases to the research findings.

Authors’ Reply: The updated manuscript integrates a new and better methodological approach. The questionnaire has been circulated again in order to increase the sample size. 

6.Only a global Cronbach’s alpha index without displaying the latent variable-specific indices can affect the data’s credibility. The presence of items with low factor loadings in the factor loading table needs to be addressed by the author to bolster the persuasiveness of the study’s outcomes.

Authors’ Reply: The statistical analysis has been revised.

7. The RMSEA value is approaching 0.1, whereas values below 0.05 are regarded as more ideal.

Authors’ Reply: The statistical analysis has been revised.

8.The robustness test is absent, has the author considered incorporating robustness tests to enhance the study’s reliability?

Authors’ Reply: In the revised manuscript the following tests are used: convergent reliability, discriminate validity, direct relationship and mediating relationship. The hypotheses have been tested using path coefficient and p values.

Above all, many parts of this paper are lack of detailed and necessary discussion, it is recommended to reject the paper.

The authors thank the reviewers for their detailed suggestions and hope that this step will make the manuscript closer to publication and help in the contribution to the research and academia.

Regards,

Hammna Jillani

hammnajillani@gmail.com

---

## [Decision Letter · Decision Letter 1]

17 Oct 2023

PONE-D-23-17774R1The mediating role of stakeholders on green banking practices and bank’s performance: the case of a developing nationPLOS ONE

Dear Dr. Jillani,

Thank you for submitting your manuscript to PLOS ONE. After careful consideration, we feel that it has merit but does not fully meet PLOS ONE’s publication criteria as it currently stands. Therefore, we invite you to submit a revised version of the manuscript that addresses the points raised during the review process.

We look forward to receiving your revised manuscript.

Kind regards,

Muhammad Hashim, PhD

Academic Editor

PLOS ONE

Journal Requirements:

Additional Editor Comments:

Managerial section need to improve significantly, merge the section Practical Implications and Managerial Implications in one heading may be "Managerial Implication" and the recommendations should be in a summaries form and preferable in points and discuss as a guideline for the stakeholders. The author get proof reading of the paper from experts. 

Reviewers' comments:

Reviewer's Responses to Questions

**Comments to the Author**

1. If the authors have adequately addressed your comments raised in a previous round of review and you feel that this manuscript is now acceptable for publication, you may indicate that here to bypass the “Comments to the Author” section, enter your conflict of interest statement in the “Confidential to Editor” section, and submit your "Accept" recommendation.

Reviewer #1: (No Response)

Reviewer #2: (No Response)

2. Is the manuscript technically sound, and do the data support the conclusions?

Reviewer #1: (No Response)

Reviewer #2: Yes

3. Has the statistical analysis been performed appropriately and rigorously? 

Reviewer #1: (No Response)

Reviewer #2: No

4. Have the authors made all data underlying the findings in their manuscript fully available?

Reviewer #1: (No Response)

Reviewer #2: Yes

5. Is the manuscript presented in an intelligible fashion and written in standard English?

Reviewer #1: (No Response)

Reviewer #2: Yes

6. Review Comments to the Author

Reviewer #1: I did not see the robust tests in my 5th comment and even the authors did not reply to this comment. If robust tests are not added, readers will find it difficult to believe that the empirical results are reliable. The notes section of the tables have not been added, and without notes, how do readers know what ** *, * *, and * represent. Please make modifications according to the comments, and if you are unable to make the necessary changes, please inform us of the reason.

Reviewer #2: I have carefully reviewed your manuscript, taking into consideration the revisions made in response to the previous round of feedback. However, I have identified several issues that require further attention and improvement to ensure the quality of your research. Here are my review comments:

1.Your abstract includes the research background, research methods, and research findings but lacks a clear explanation of the research innovation or novelty. I recommend providing a clear statement of your research question and the unique contribution of your study in the abstract to capture the reader's interest.

2.In your paper, there is insufficient explanation regarding the dimensions and items selected for your variables. Please provide more detailed information, explaining the theoretical basis and rationale behind your variable selection, to help readers better understand your research design.

3.You mention four types of stakeholders-competitors, consumers, shareholders, and top management of the organization, but you only collect data from top management. I suggest that you discuss in detail the roles of the other three types of stakeholders in your research and explain why you chose not to collect data from them or provide valid reasons for this decision.

4.The absence of a table displaying model fit indices such as RMSEA, CFI, and TLI makes it impossible to assess the quality of model fit. Please include the results of these indices and explain their significance and relevance to your study.

5.The factor loading threshold you use in the paper (greater than 0.4) is lower than the typically required threshold (greater than 0.7). Please provide a theoretical basis or prior research support for your choice of this lower threshold and ensure its validity and interpretability.

6.Your third research objective is to determine the role of stakeholders in the implementing of green banking guidelines, but the paper mainly focuses on interviews with banking personnel. Please specify how you plan to represent the roles of the other three types of stakeholders or provide a valid explanation for not studying them.

7.Have you considered conducting robustness checks to enhance the reliability of your research? If so, please provide details and results of any robustness checks you have conducted.

In summary, your research has the potential to make a valuable contribution, but further revision and improvement are necessary to meet the standards of academic publishing. Please address and elaborate on the above issues in your revised manuscript to facilitate the continuation of the review process.

I wish you success in completing your revision work and look forward to seeing your final manuscript.

7. PLOS authors have the option to publish the peer review history of their article (what does this mean?). If published, this will include your full peer review and any attached files.

Reviewer #1: No

Reviewer #2: No

---

## [Author Response · Author response to Decision Letter 1]

1 Dec 2023

1.0 Reviewer #1

Authors’ Reply: Thanks for the valuable feedback. The robustness test has been added in heading 4.3. Moreover, the notes for the required tables have been added. 

Reviewer #1: I did not see the robust tests in my 5th comment and even the authors did not reply to this comment. If robust tests are not added, readers will find it difficult to believe that the empirical results are reliable. The notes section of the tables have not been added, and without notes, how do readers know what ** *, * *, and * represent. Please make modifications according to the comments, and if you are unable to make the necessary changes, please inform us of the reason.

2.0 Reviewer #2

Reviewer #2: I have carefully reviewed your manuscript, taking into consideration the revisions made in response to the previous round of feedback. However, I have identified several issues that require further attention and improvement to ensure the quality of your research. Here are my review comments:

1.Your abstract includes the research background, research methods, and research findings but lacks a clear explanation of the research innovation or novelty. I recommend providing a clear statement of your research question and the unique contribution of your study in the abstract to capture the reader's interest.

Authors’ Reply: Abstract has been updated ‘This research fills the gap in existing literature by testing and implying the mediating role of Stakeholders’ Influence on the relationship between Green Banking Practices and the Bank’s Performance.’

2.In your paper, there is insufficient explanation regarding the dimensions and items selected for your variables. Please provide more detailed information, explaining the theoretical basis and rationale behind your variable selection, to help readers better understand your research design.

Authors’ Reply: The items and variables are adapted form the following sources. ‘A structured questionnaire was used to obtain primary data from bankers from various banks [19,37,40,54]’

3.You mention four types of stakeholders-competitors, consumers, shareholders, and top management of the organization, but you only collect data from top management. I suggest that you discuss in detail the roles of the other three types of stakeholders in your research and explain why you chose not to collect data from them or provide valid reasons for this decision.

Authors’ Reply: The authors take this comment as valuable in designing their future research. We believe that we can do further research and discuss the role of each stakeholder in detail. For this work the general role of investigated using the viewpoint of the banking personnel. 

4.The absence of a table displaying model fit indices such as RMSEA, CFI, and TLI makes it impossible to assess the quality of model fit. Please include the results of these indices and explain their significance and relevance to your study.

Authors’ Reply: CFA table has been added under the heading 4.4 and all the values have been provided. 

5.The factor loading threshold you use in the paper (greater than 0.4) is lower than the typically required threshold (greater than 0.7). Please provide a theoretical basis or prior research support for your choice of this lower threshold and ensure its validity and interpretability.

Authors’ Reply: The individual values of 0.4 have not been dropped as the collective sum of cronbach’s alpha using these item loading was more than 0.6. The items used to represent constructs pose satisfactory internal consistent relationships and a Cronbach’s alpha value of i.e. >0.5 [58]. The low factor loadings collectively gave a good value of Cronbach’s alpha therefore, items are not dropped. Reference provided. 

6.Your third research objective is to determine the role of stakeholders in the implementing of green banking guidelines, but the paper mainly focuses on interviews with banking personnel. Please specify how you plan to represent the roles of the other three types of stakeholders or provide a valid explanation for not studying them.

Authors’ Reply: The research aims to identify the role of stakeholders from the viewpoint of banking personnel . Their opinions are incorporated to check whether stakeholders are an important component in implementation of GB and have a role in improving Bank’s performance or not. 

7.Have you considered conducting robustness checks to enhance the reliability of your research? If so, please provide details and results of any robustness checks you have conducted.

Authors’ Reply: The robustness test has been added in heading 4.3

In summary, your research has the potential to make a valuable contribution, but further revision and improvement are necessary to meet the standards of academic publishing. Please address and elaborate on the above issues in your revised manuscript to facilitate the continuation of the review process.

I wish you success in completing your revision work and look forward to seeing your final manuscript.

The authors thank the reviewers for their detailed suggestions and hope that this step will make the manuscript closer to publication and help in the contribution to the research and academia.

---

## [Decision Letter · Decision Letter 2]

2 Feb 2024

PONE-D-23-17774R2The mediating role of stakeholders on green banking practices and bank’s performance: the case of a developing nationPLOS ONE

Dear Dr. Jillani,

Thank you for submitting your manuscript to PLOS ONE. After careful consideration, we feel that it has merit but does not fully meet PLOS ONE’s publication criteria as it currently stands. Therefore, we invite you to submit a revised version of the manuscript that addresses the points raised during the review process.

We look forward to receiving your revised manuscript.

Kind regards,

Muhammad Hashim, PhD

Academic Editor

PLOS ONE

Reviewers' comments:

Reviewer's Responses to Questions

**Comments to the Author**

1. If the authors have adequately addressed your comments raised in a previous round of review and you feel that this manuscript is now acceptable for publication, you may indicate that here to bypass the “Comments to the Author” section, enter your conflict of interest statement in the “Confidential to Editor” section, and submit your "Accept" recommendation.

Reviewer #1: (No Response)

Reviewer #2: (No Response)

Reviewer #3: (No Response)

2. Is the manuscript technically sound, and do the data support the conclusions?

Reviewer #1: (No Response)

Reviewer #2: Partly

Reviewer #3: (No Response)

3. Has the statistical analysis been performed appropriately and rigorously? 

Reviewer #1: (No Response)

Reviewer #2: No

Reviewer #3: (No Response)

4. Have the authors made all data underlying the findings in their manuscript fully available?

Reviewer #1: (No Response)

Reviewer #2: Yes

Reviewer #3: (No Response)

5. Is the manuscript presented in an intelligible fashion and written in standard English?

Reviewer #1: (No Response)

Reviewer #2: No

Reviewer #3: (No Response)

6. Review Comments to the Author

Reviewer #1: The authors need to pay attention to the design of tables and the figure. For example, Figure 1, which is not very clear if directly used the original figure of the software.

Reviewer #2: I have carefully reviewed the revised version of the manuscript. Although the manuscript is improved, it have not addressed the issues that I have mentioned before. And the revised version is confused to read, it is unclear which are revised compared with the previous version. In summary, I think this manuscript have not meet the standards of academic publishing, and recommend to reject it.

Reviewer #3: The idea of the paper is very good and in accordance with current market hot issues. However, here some issues are identified, If the author incorporate these points then this paper can be accepted.

1.There is a weak linkage between different parts of the paper. Particularly in the literature review, the author introduces four stakeholder groups, namely competitors, consumers, shareholders, and top organizational executives. While the subsequent analysis only involves the government and banks.

2. The hypotheses are given without any references or theoretical backgrounds so there must be sound theoretical linkage with hypothesis.

3. The language of the introduction section is poor grammatically so it should be improved.

4.The sample size of the paper is close to the lower limit of the SEM model’s requirements. An insufficient sample can result in biases to the research findings. So, it should be increased.

5.Only a global Cronbach’s alpha index without displaying the latent variable-specific indices can affect the data’s credibility. The presence of items with low factor loadings in the factor loading table needs to be addressed by the author to bolster the persuasiveness of the study’s outcomes.

6.The robustness test is absent, has the author considered incorporating robustness tests to enhance the study’s reliability? If not u should add in the study otherwise there will be question mark on the credibility of the results.

7. PLOS authors have the option to publish the peer review history of their article (what does this mean?). If published, this will include your full peer review and any attached files.

Reviewer #1: No

Reviewer #2: No

Reviewer #3: **Yes: **Hafiz Ahmad Ashraf

---

## [Author Response · Author response to Decision Letter 2]

7 Feb 2024

Dear Editor,

PLOS One.

Subject: Rebuttal Letter

Please be informed that the authors acknowledge the time-taking and guiding suggestions of the reviewers and the editor. We believe that the valuable comments can make the paper more effective in terms of publication and readiness. We have meticulously reviewed and addressed each comment, and all modifications have been highlighted in the revised manuscript for ease of reference. We humbly request your reconsideration of our work and eagerly await your further feedback. The required documents are attached to the submission. 

Reviewer #1: The authors need to pay attention to the design of tables and the figure. For example, Figure 1, which is not very clear if directly used the original figure of the software.

Authors reply: The size ratio has been increased.

Reviewer #2: I have carefully reviewed the revised version of the manuscript. Although the manuscript is improved, it have not addressed the issues that I have mentioned before. And the revised version is confused to read, it is unclear which are revised compared with the previous version. In summary, I think this manuscript have not meet the standards of academic publishing, and recommend to reject it.

Reviewer #3: The idea of the paper is very good and in accordance with current market hot issues. However, here some issues are identified, If the author incorporate these points then this paper can be accepted.

1.There is a weak linkage between different parts of the paper. Particularly in the literature review, the author introduces four stakeholder groups, namely competitors, consumers, shareholders, and top organizational executives. While the subsequent analysis only involves the government and banks.

Authors reply: The perspective of bankers is used to identify the role of stakeholders. The analysis focuses on the bankers; perspective only. An individual analysis of the stakeholders can be done as a separate study.

2. The hypotheses are given without any references or theoretical backgrounds so there must be sound theoretical linkage with hypothesis.

Authors reply: The references for hypotheses have been added.

2.4.1 Green Banking and Bank’s Performance

Shaumya and Arulrajah [37] found a significant and positive relation between green banking practices on the environmental performance of the bank in Sri Lanka, thus indicating that the performance is increased if green practices are followed. In short, all the activities should be combined to foster the environmental growth of a bank. The bank’s employee-related, customer-related, operational, and policy-related activities have a direct influence on the implementation of green banking policy and green financing thus improving the environmental performance of the bank. Moreover, energy-efficient equipment and a well-designed environmental policy along with employee training and awareness sessions also contribute to the bank’s environmental performance. In Bangladesh, green funding is promoted to expand the environmental performance of the banks consequently leading to the economic development of the country [38, 39]. Choudhury et al., [40] suggest that greening the banking sector and taking an environmentally proactive approach can result in functional improvements and operational efficiencies in the banking sector. Furthermore, the bank’s image can be enhanced if green practices are adopted [41,42]. The literature backs that adopting green banking practices can lead to the financial, operational, and environmental performance of the bank [43]. Henceforth, the following hypothesis is proposed.

H1: There is a significant relationship between Green banking practices and the Bank’s performance.

2.4.2 Stakeholders and Bank’s Performance

 In the 20th century, stakeholders started recognizing that environmental degradation and natural resource degeneration are the greatest externalities being produced by organizations, through operation and business activities. This resulted in increasing stakeholder pressure on organizations to reduce their adverse environmental impacts. Due to the formation of environmental conferences and international protocols, excessive pressure through external groups was formed to conserve the environment and incorporate sustainability in business operations [16]. In the beginning, the banking sector was not included in the organizations that harm the environment directly and require moderation in the policies and procedures but later on, the indirect impacts of the banking sector were identified and this sustainable finance and banking came into action. 

For the adoption of green banking policy, pressure from all the stakeholder groups and international organizations has a direct influence. Moreover, social pressure plays a chief role in attaining sustainable growth and development of an organization. According to the stakeholder's theory, all the groups must be equally involved, not just the financers, to make the system grow successfully and work efficiently [44, 45]. The Stakeholder Theory projects that an organization occurs for the profit and benefit of numerous stakeholders moreover it produces externalities through its business activities that can affect stakeholders [46]. As a consequence of these externalities, there is the proliferation of stakeholder pressures on firms to shrink their negative impacts. Similar to the Stakeholder Theory, the Institutional Theory also supports the stakeholder approach by arguing that creating stakeholder engagement has become essential for organizations to establish social acceptability and competitiveness along with social sustainability [47]. 

 There exist many groups of stakeholders including media, special interest groups, employees, research community government, etc. but in the literature, four major stakeholder groups are identified that have a direct impact and influence the speeding up of the development process in any organization. The four main groups are competitors, consumers, stockholders, and top management of the organization. All the shareholders play an important role in the development of green banking procedures and activities and stakeholders should be kept on the same page through effective communication by the management of any bank [40]. 

According to a study by Mehedi et al., [48], the organizational pressure of various stakeholders and environmental policy has the highest influence on any organization to develop and improve sustainability in business. Shafique and Majeed [41], explored the bankers’ intention to adopt green banking which can be influenced by Policy Guidelines, Attitude towards usage, Central Bank Regulations, and Management commitment and support. Linh and Anh [49], used a mixed methods approach of the questionnaire and in-depth interviews by the bank officials and the findings show the immense importance of stakeholders on green banking. Moreover, the study highlighted various benefits as green banking improves ties between local and international organizations, provides new opportunities for business growth thus minimizing capital losses, and engages all the stakeholders for organizational growth. This leads to the development of the following hypothesis.

H2: There is a significant relationship between Green banking practices and Stakeholders’ Influence.

H3: Stakeholders’ influence mediates the relationship between Green banking practices and the Bank’s performance. 

3. The language of the introduction section is poor grammatically so it should be improved.

Authors reply: The introduction has been revised.

4.The sample size of the paper is close to the lower limit of the SEM model’s requirements. An insufficient sample can result in biases to the research findings. So, it should be increased.

Authors reply: The sample size has been used as per reference paper. 

References: 19,37,40,54

5.Only a global Cronbach’s alpha index without displaying the latent variable-specific indices can affect the data’s credibility. The presence of items with low factor loadings in the factor loading table needs to be addressed by the author to bolster the persuasiveness of the study’s outcomes.

Authors reply: The low factor loading values when combined give a good Cronbach’s alpha value, therefore retained.

The items used to represent constructs pose satisfactory internal consistent relationships and a Cronbach’s alpha value of i.e. >0.5 [58]. The low factor loadings collectively gave a good value of Cronbach’s alpha therefore, items are not dropped.

6.The robustness test is absent, has the author considered incorporating robustness tests to enhance the study’s reliability? If not u should add in the study otherwise there will be question mark on the credibility of the results.

Authors reply: The robustness test has been added.

4.3 Assessment of Non-Linearity

Table 4 indicates the result of Ramsey’s RESET to assess the non-linearity. It is indicated that there exists no partial regression of BP on SI and GB. The results of bootstrapping indicate that neither of the nonlinear effects is significant thus indicating that the linear effects model is robust.

Non- Linear Relationship Coefficient p value f Ramsey’s RESET

GB*GB-> SI 0.051 0.387 0.002 F(3,242)= 0.97, p=0.598

GB*GB -> BP 0.035 0.490 0.001 

SI*SI -> BP 0.023 0.448 0.001 

Table 4: Non-Linearity Test

The authors thank the reviewers for their detailed suggestions and hope that this step will make the manuscript closer to publication and help in the contribution to the research and academia.

Regards,

Hammna Jillani

hammnajillani@gmail.com

---

## [Decision Letter · Decision Letter 3]

1 Mar 2024

The mediating role of stakeholders on green banking practices and bank’s performance: the case of a developing nation

PONE-D-23-17774R3

Dear Dr. Jillani,

We’re pleased to inform you that your manuscript has been judged scientifically suitable for publication and will be formally accepted for publication once it meets all outstanding technical requirements.

Kind regards,

Muhammad Hashim, PhD

Academic Editor

PLOS ONE
---

## [Editor Report · Acceptance letter]

29 Apr 2024

PONE-D-23-17774R3 

PLOS ONE

Dear Dr. Jillani, 

I'm pleased to inform you that your manuscript has been deemed suitable for publication in PLOS ONE. Congratulations! Your manuscript is now being handed over to our production team.

Kind regards, 

on behalf of

Dr. Muhammad Hashim 

Academic Editor

PLOS ONE